# Non-Singular Model of Magnetized Black Hole Based on Nonlinear Electrodynamics

**Sergey I. Kruglov [1,2]**

1   Department of Physics, University of Toronto, 60 St. Georges St., Toronto, ON M5S 1A7, Canada;
    serguei.krouglov@utoronto.ca
2   Department of Chemical and Physical Sciences, University of Toronto Mississauga, 3359 Mississauga Road
    North, Mississauga, ON L5L 1C6, Canada

**Abstract:** A new modified Hayward metric of magnetically charged non-singular black hole spacetime in the framework of nonlinear electrodynamics is constructed. When the fundamental length introduced, characterising quantum gravity effects, vanishes, one comes to the general relativity coupled with the Bronnikov model of nonlinear electrodynamics. The metric can have one (an extreme) horizon, two horizons of black holes, or no horizons corresponding to the particle-like solution. Corrections to the Reissner–Nordström solution are found as the radius approaches infinity. As $r \to 0$ the metric has a de Sitter core showing the absence of singularities, the asymptotic of the Ricci and Kretschmann scalars are obtained and they are finite everywhere. The thermodynamics of black holes, by calculating the Hawking temperature and the heat capacity, is studied. It is demonstrated that phase transitions take place when the Hawking temperature possesses the maximum. Black holes are thermodynamically stable at some range of parameters.

**Keywords:** modified Hayward metric; magnetically charged black hole; nonlinear electrodynamics; thermodynamics

---

## 1. Introduction

It is well-known that General Relativity (GR) is ultraviolet (UV) incomplete. In addition, there is a problem of singularities in the classical Einstein theory of gravity. Thus, solutions of the Einstein equations for charged (the Reisner–Nordström metric) black holes (BHs) have curvature singularities in the center ($r = 0$). Therefore, GR should be modified when the curvature is large. There are some attempts to overcome problems in the classical Einstein theory of gravity. So, if one adds curvature terms of the higher order or terms with higher derivatives, the UV behaviour of the Einstein gravity will be improved [1,2]. But the price for this is the existence of ghosts (non-physical degrees of freedom). A ghost free modification of the GR, which is UV-complete, was considered in References [3–5] but such a theory is non-local and has an infinite number of derivatives. Because the fundamental quantum gravity theory (UV-complete) is absent, some phenomenological models can be useful for solving problems of singularities. Following References [6–8] (see also Reference [9]), we assume that there is a critical energy $\mu$ and the corresponding length $l = \mu^{-1}$ in such a way that the metric is modified when the spacetime curvature is in the order of $l^{-2}$. The length scale, characterising quantum gravity effects, is smaller than $l$ and one may use the classical metric $g_{\mu\nu}$. In addition, we suppose that the limiting curvature condition $\mathcal{R} \leq cl^{-2}$ ($\mathcal{R}$ is one of the curvature invariants, $c$ is dimensionless constant depending on the curvature invariant) is satisfied [6–8]. A simple metric satisfying the above conditions was proposed by Hayward [10] for a neutral BH. This is the phenomenological approach that we explore here.

The first pioneering work representing a regular BH in GR is in Reference [11]. It was shown in Reference [12] that the Bardeen model can describe the gravitational field of a nonlinear magnetic monopole. In References [13,14] the regular electrically charged BH solution in GR was presented, where the source is a nonlinear electrodynamics (NED) field satisfying the weak energy condition. It worth noting that in accordance with Reference [15], regular electric solutions with the Maxwell weak-field limit can be described only by different NED theories in different parts of spacetime. Thus, there is a significant shortcoming in the models of References [13,14].

In this paper, we consider the spherically symmetric non-singular model of the magnetically charged BH based on NED. In some NED, the electric field in the center of point-like charges is finite [16–20] and the self-energy of charges is finite unlike classical electrodynamics. It is worth mentioning that quantum corrections to Maxwell's electrodynamics, within QED , lead to NED [21]. The universe inflation also can be explained in the framework of the GR coupled with NED [22–29].

Here, with the help of the modified Hayward metric, we study regular magnetically charged BH solutions within NED considered in Reference [15]. The BH thermodynamics and phase transitions are investigated. In References [30,31] the authors also considered BH solutions with the modified Hayward metric based on NED proposed in References [32,33], respectively. The thermodynamics for a magnetically charged regular BH, which comes from the action of GR and NED, was investigated in Reference [34]. These authors also used NED proposed in Reference [15]. A similar study was performed in References [35,36]. The work in Reference [37] analyzes the minimal model proposed by Hayward for an uncharged BH within GR. The authors introduced an anisotropic fluid and postulated the expressions for the energy density and pressure but the Lagrangian corresponding to the stress tensor was not obtained. In the present study we use the NED of Reference [15], explore a phenomenological extension of GR by introducing a fundamental length $l$ using the modified Hayward metric, and investigate the magnetically charged BH.

This paper is organized as follows. In Section 2 the modified Hayward metric is studied and we obtain the asymptotic of the metric and mass functions as $r \to 0$ and $r \to \infty$. Corrections to the Reissner–Nordström (RN) solution are found. The asymptotic of the Ricci and Kretschmann scalars are calculated and we show that curvature singularities are absent. In Section 3 we calculate the Hawking temperature and the heat capacity of BHs. We demonstrate that the second-order phase transitions occur. It is shown that in some range of parameters BHs are stable. Section 4 is a conclusion.

## 2. A Regular Magnetized BH Solution

To describe the magnetically charged BH solution we consider the Lagrangian density of NED [15]:

$$\mathcal{L} = -\frac{\mathcal{F}}{\cosh^2 \sqrt[4]{|\beta \mathcal{F}|}}, \tag{1}$$

where $\mathcal{F} = (1/4)F_{\mu\nu}F^{\mu\nu} = (\mathbf{B}^2 - \mathbf{E}^2)/2$ and the field tensor is $F_{\mu\nu} = \partial_\mu A_\nu - \partial_\nu A_\mu$. The parameter $\beta$ in Equation (1) is positive and it possesses the dimension of (length)$^4$. At the weak field limit the Lagrangian density (1) becomes:

$$\mathcal{L} \to -\mathcal{F} \quad \beta \mathcal{F} \ll 1, \tag{2}$$

that is, the correspondence principle holds. We will derive the metric function representing the static magnetic regular BH. Let us consider the spherically symmetric line element, which is given by:

$$ds^2 = -f(r)dt^2 + \frac{1}{f(r)}dr^2 + r^2(d\vartheta^2 + \sin^2 \vartheta d\phi^2). \tag{3}$$

The Hayward metric function [10] is given by:

$$f(r) = 1 - \frac{2GMr^2}{r^3 + 2GMl^2}, \tag{4}$$

where $G$ is the Newton constant, $M$ = constant and $l$ is the fundamental length. We interpret this metric in the framework of an extension of GR for an uncharged source and replace the Schwarzschild metric. It should be noted that in GR the metric function (4) may be obtained as a solution within NED with a nonzero magnetic charge [38]. But in this case the NED Lagrangian is ill-defined. At the weak-field limit the NED Lagrangian does not approach to the Maxwell Lagrangian. One can consider and investigate other geometries of the horizon in Equation (3). At $l = 0$ we come to the Schwarzschild metric of a BH which is a solution to Einstein's equation without sources. Now we suppose that the BH is magnetically charged. Then the mass function of a BH varies with $r$ and is:

$$M(r) = m_0 + \int_0^r \rho(r) r^2 dr = m_0 + \int_0^\infty \rho(r) r^2 dr - \int_r^\infty \rho(r) r^2 dr, \tag{5}$$

where $m_0$ is the Schwarzschild mass, $\rho(r)$ is the magnetic energy density and $m_M = \int_0^\infty \rho(r) r^2 dr$ is the magnetic mass of the BH. In Reference [15] the mass $m_0$, which can be considered as a constant of integration, was not introduced. But the case $m_0 \neq 0$ allows us to consider the uncharged BH when the charge $q = 0$. Indeed, if $q = 0$ ($\rho(r) = 0$) in Equation (5), the mass function $M$ becomes constant and we come to the Hayward metric function (4). At $\mathbf{E} = 0$ the magnetic energy density, corresponding to Equation (1), is given by:

$$\rho(r) = -\mathcal{L} = \frac{\mathcal{F}}{\cosh^2 \sqrt[4]{|\beta \mathcal{F}|}}, \tag{6}$$

where $\mathcal{F} = B^2/2 = q^2/(2r^4)$, and $q$ is a magnetic charge. Then the mass function (5) becomes:

$$M(r) = m_0 + m_M - \frac{q^{3/2}}{2^{3/4} \beta^{1/4}} \tanh\left(\frac{\beta^{1/4} \sqrt{q}}{2^{1/4} r}\right), \tag{7}$$

where the BH magnetic mass is given by [36]:

$$m_M = \int_0^\infty \rho(r) r^2 dr = \frac{q^{3/2}}{2^{3/4} \beta^{1/4}}. \tag{8}$$

The total BH mass is $M(\infty) \equiv m = m_0 + m_M$. Then the metric function corresponding to a charged BH is:

$$f(r) = 1 - \frac{2GM(r) r^2}{r^3 + 2GM(r) l^2}, \tag{9}$$

where $M(r)$ is given by (7).

For a convenience we introduce the dimensionless parameter $x = 2^{1/4} r / (\beta^{1/4} \sqrt{q})$. Then from Equations (7)–(9) one obtains the metric function:

$$f(x) = 1 - \frac{Ax^2 g(x)}{x^3 + Bg(x)}, \tag{10}$$

where:

$$A = \frac{\sqrt{2} Gq}{\sqrt{\beta}}, \quad B = \frac{2Gl^2}{\beta}, \quad C = \frac{2^{3/4} \beta^{1/4} m_0}{q^{3/2}}, \quad g(x) = C + 1 - \tanh\left(\frac{1}{x}\right). \tag{11}$$

From Equations (10) and (11) we find the asymptotic of the metric function as $r \to \infty$ and $r \to 0$:

$$f(r) = 1 - \frac{2Gm}{r} + \frac{Gq^2}{r^2} - \frac{G}{r^4}\left(\frac{\sqrt{\beta} q^3}{3\sqrt{2}} - 4Gl^2 m^2\right) + \mathcal{O}(r^{-5}) \quad r \to \infty, \tag{12}$$

$$f(r) = 1 - \frac{r^2}{l^2} + \frac{r^5}{2Gm_0 l^4} + \mathcal{O}(r^6) \quad r \to 0. \tag{13}$$

Equation (12) shows the corrections to the RN solution that are in the order of $\mathcal{O}(r^{-4})$. At $l = 0$ and $m_0 = 0$ (when the total BH mass is the magnetic mass) Equation (12) is converted into the equation obtained in Reference [36]. As $r \to \infty$ we have $f(\infty) = 1$, and the spacetime becomes flat. According to Equation (13) $\lim_{r \to 0} f(r) = 1$. Thus, the spacetime has a smooth de Sitter core and the BH is regular. If $\beta = 0, l = 0$ one has the RN solution. The plot of the function $f(x)$ is depicted in Figure 1. This plot is typical for any regular solution described by the metric (3) (see, e.g., Reference [13]).

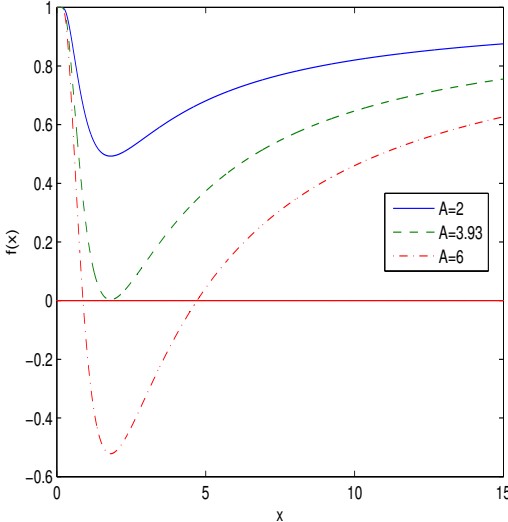

**Figure 1.** The plot of the function $f(x)$ for $B = 1$ and $C = 0$ ($m_0 = 0$). The dashed-dotted line corresponds to $A = 6$, the solid line corresponds to $A = 2$ and the dashed line corresponds to $A = 3.93$.

In accordance with Figure 1 at $A < 3.93$ ($B = 1, C = 0$) horizons are absent and we have particle-like solution. At $A \approx 3.93$ the horizons shrink into one horizon (the extreme solution). If $> 3.93$, we have two horizons of a BH. The horizon radii $x_h$ are roots of the equation $f(x_h) = 0$. From Equation (10), at $B = 1, C = 0$ one finds the inner $x_-$ ($x_- = 2^{1/4} r_- / (\beta^{1/4} \sqrt{q})$) and outer $x_+$ ($x_+ = 2^{1/4} r_+ / (\beta^{1/4} \sqrt{q})$) horizon radii of the BH that are given in Table 1.

**Table 1.** The BH inner and outer horizon radii ($B = 1, C = 0$).

| $A$ | 4 | 5 | 6 | 7 | 8 | 9 | 10 | 15 |
|-----|------|------|------|------|------|------|------|-------|
| $x_-$ | 1.56 | 1.04 | 0.88 | 0.78 | 0.72 | 0.68 | 0.64 | 0.54 |
| $x_+$ | 2.11 | 3.58 | 4.71 | 5.78 | 6.82 | 7.84 | 8.87 | 13.92 |

The asymptotic of the Ricci and Kretschmann scalars can be obtained from the relations:

$$R = -f''(r) - \frac{4}{r}f'(r) - 2\frac{f(r) - 1}{r^2}, \tag{14}$$

$$K = f''^2(r) + \left(\frac{2f'(r)}{r}\right)^2 + \frac{4(f(r) - 1)^2}{r^4}. \tag{15}$$

From Equations (14) and (15) we find:

$$R = \frac{12}{l^2} - \frac{21r^3}{Gm_0 l^4} + \mathcal{O}(r^4) \quad r \to 0, \tag{16}$$

$$R = -\frac{86Gm}{r^3} + \frac{12Gq^2}{r^4} - \frac{34G}{r^6}\left(\frac{\sqrt{\beta}q^3}{3\sqrt{2}} - 4Gl^2m^2\right) + \mathcal{O}(r^{-7}) \quad r \to \infty, \tag{17}$$

$$K = \frac{24}{l^4} - \frac{84r^3}{Gm_0l^6} + \mathcal{O}(r^4) \quad r \to 0, \tag{18}$$

$$K = \frac{48G^2m^2}{r^6} - \frac{96G^2mq^2}{r^7} + \mathcal{O}(r^{-8}) \quad r \to \infty. \tag{19}$$

As $r \to \infty$ the Ricci and Kretschmann scalars vanish and the spacetime becomes flat. Equations (16)–(19) indicate that solutions obtained are regular.

## 3. Thermodynamics and Phase Transitions

Let us study the thermal stability of magnetized BHs and the possible phase transitions. The Hawking temperature is given by [39]:

$$T_H = \frac{\kappa}{2\pi} = \frac{f'(r_h)}{4\pi}, \tag{20}$$

where $\kappa$ is the surface gravity and $r_h$ is the horizon radius. Making use of Equations (10) and (20) we obtain the Hawking temperature:

$$T_H = \frac{\sqrt{q}G}{2^{5/4}\pi\beta^{3/4}(x_h^3 + Bg(x_h))}\left(-2x_h g(x_h)\right.$$

$$\left. - \frac{1}{\cosh^2(1/x_h)} + \frac{g(x_h)(B + 3x_h^4\cosh^2(1/x_h))}{(x_h^3 + Bg(x_h))\cosh^2(1/x_h)}\right). \tag{21}$$

The horizon radii $r_h$ (and $x_h$) are defined as roots of the equation $f(r_h) = 0$ (and $f(x_h) = 0$). From Equation (10) we obtain:

$$\frac{Gq}{\sqrt{\beta}} = \frac{x_h^3 + Bg(x_h)}{\sqrt{2}x_h^2 g(x_h)}. \tag{22}$$

According to Equation (22) the horizon radius $r_h$ (and $x_h$) depends on the magnetic charge $q$ and the model parameter $\beta$. Substituting $Gq/\sqrt{\beta}$ from Equation (22) into Equation (21) we obtain the final equation for the Hawking temperature:

$$T_H = \frac{1}{2^{7/4}\pi\sqrt{q}\beta^{1/4}g(x_h)x_h^2}\left(-2x_h g(x_h)\right.$$

$$\left. - \frac{1}{\cosh^2(1/x_h)} + \frac{g(x_h)(B + 3x_h^4\cosh^2(1/x_h))}{(x_h^3 + Bg(x_h))\cosh^2(1/x_h)}\right). \tag{23}$$

The plot of the reduced Hawking temperature $T_H\sqrt{q}\beta^{1/4}$ is depicted in Figure 2 for different values of the parameter $B$ (or $l$).

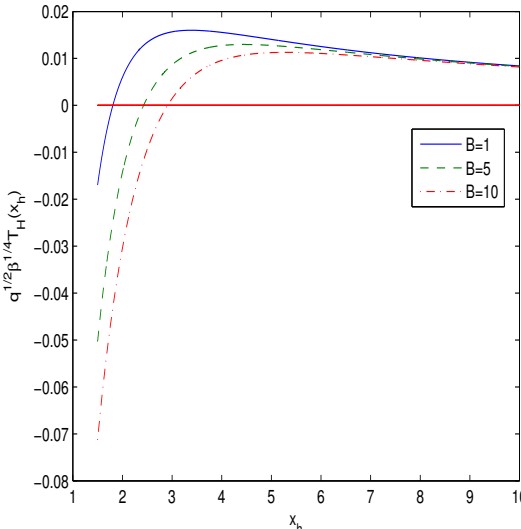

**Figure 2.** The plot of the function $T_H\sqrt{q}\beta^{1/4}$ vs. horizons $x_h$ for $C = 0$ ($m_0 = 0$). The dashed-dotted line corresponds to $B = 10$, the solid line corresponds to $B = 1$ and the dashed line corresponds to $B = 5$.

It follows from Figure 2 that at the bigger value of $l$ (or $B$) the maximum of the Hawking temperature shifts to the bigger value of the horizon radius. The temperature curve has one extremum (maximum) resulting in one phase transition during the evaporation. Similar form of the temperature curve for a BH takes place in the models studied in [36]. The heat capacity at the constant charge is defined by the relation [40]:

$$C_q = T_H \left( \frac{\partial S}{\partial T_H} \right)_q = \frac{T_H \partial S/\partial r_h}{\partial T_H/\partial r_h} = \frac{2\pi r_h T_H}{G \partial T_H/\partial r_h}. \tag{24}$$

The entropy obeys the Hawking area low $S = A/(4G) = \pi r_h^2/G$. When the Hawking temperature has the extremum ($\partial T_H/\partial r_h = 0$) the heat capacity is singular and the second-order phase transition takes place. In Figure 3 the function $GC_q/(\sqrt{\beta}q)$ vs. the horizon radius $x_h$ for different values of $B$ for $C = 0$ ($m_0 = 0$) is presented.

Figure 2 shows the similarity in the considered thermodynamics of our model and the thermodynamics of a neutral regular BH. It is seen from Figure 3 that second-order phase transitions at the discontinuity points occur between negative and positive heat capacities. Figure 2 shows that the maximum of the temperature (the Davies point), where the phase transitions take place, separates areas with increasing and decreasing BH temperatures. The unstable point between the positive and negative heat capacities has a discontinuity. The positive heat capacity corresponds to the late stage and the negative capacity to the early stage of the thermodynamics process. Thus, there is an interval of the horizon radius where the heat capacity is positive and the BH is stable. In accordance with Figures 2 and 3, the heat capacity possesses a discontinuity at the horizon where the Hawking temperature possesses a maximum. When the parameter $B$ is bigger, the second-order phase transition of the BH occurs at the larger value of the horizon radius $r_h$ ($x_h$). For the large value of $x_h$ the BH is unstable ($C_q < 0$).

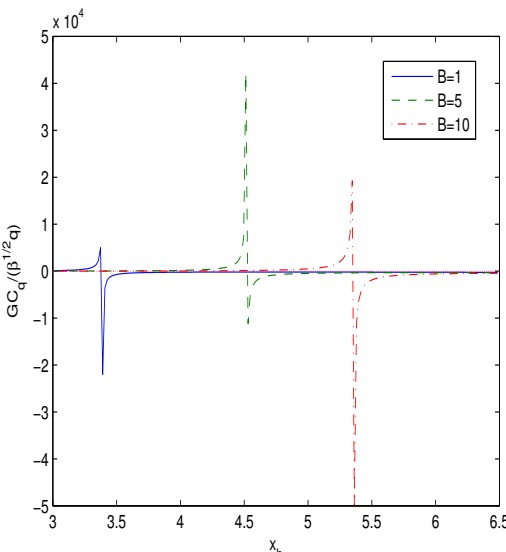

**Figure 3.** The plot of the function $C_q G / (\sqrt{\beta} q)$ vs. $x_h$ for $C = 0$ ($m_0 = 0$). The dashed-dotted line corresponds to $B = 10$, the solid line corresponds to $B = 1$ and the dashed line corresponds to $B = 5$.

## 4. Conclusions

Solutions of a magnetically charged regular BH in the new model were obtained. This model is of interest because of its simplicity. We found the mass and metric functions possessing simple analytical structures. The BH can have one (an extreme horizon), two horizons (trapping horizons), or no horizons (untrapped surface, see Figure 1). These plots are typical for any regular solution described by the metric (3). One can find the same behavior of the metric function in Reference [13] for another BH model. Corrections to the RN solution that are in the order of $\mathcal{O}(r^{-4})$ were obtained as the radius approaches to infinity. As $r \to \infty$ the spacetime becomes flat. The model of a electrically charged BH [13] was formulated in so-called *P*-frame (the Hamiltonian framework). But the Lagrangian dynamics is specified in *F*-framework. It was shown in Reference [15] that the regular electric solution in *P*-frame corresponds to different Lagrangians in different parts of the space if the function $P(F)$ is not monotonic. But this problem is absent for magnetic solutions. Thus, in the model [13] the problem of singularities was not solved completely [41]. We calculated the asymptotic of the Ricci and Kretschmann scalars as $r \to \infty$ and $r \to 0$ showing the absence of singularities. It was shown that the spacetime as $r \to 0$ has a de Sitter core (the flatness at the center). Thus, the singularity at $r = 0$ has been smoothed out. Our solution describes nonsingular BH with the finite curvature everywhere including $r = 0$. The regular behavior of the Ricci and Kretschmann scalars also was observed in Reference [13]. The Hawking temperature and the heat capacity of the BH were found demonstrating that second-order phase transitions take place. It was shown that second-order phase transitions separate areas between negative and positive heat capacities and areas with increasing and decreasing BH temperatures. For small values of the horizon radius, depending on the parameters of the model, the Hawking temperature is negative (see Figure 2). The thermodynamic stabilities of black holes were studied and it was shown that in some range of horizon radii the BHs are stable (the heat capacity is positive) (see Figure 3). The long standing problem of singularities is solved in this model, at $r = 0$ curvature invariants are finite and the BH is regular. In addition, at the weak field limit NED (1) becomes the Maxwell electrodynamics, that is, the correspondence principle holds. It is worth noting that, in the Bardeen model, the correspondence principle breaks out [12].

**Funding:** This research received no external funding.

**Conflicts of Interest:** The authors declare no conflict of interest.

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
