# Peer review of "Non-Singular Model of Magnetized Black Hole Based on Nonlinear Electrodynamics"

_universe, doi:10.3390/universe5120225_

Round 1

Reviewer 1 Report

This paper produces rather a strange impression. The author claims that he presents "a magnetically charged regular BH in modified theory of gravity coupled with NED", but then this must be reflected in the title. On the other hand, the author is using the NED [20] specified by the Lagrangian (2), which, being coupled to GR, gives the metric (3) with the mass function (7). However, instead, it is said that f(r) is taken in Hayward's form (3) which corresponds to a solution of a modified theory of gravity. But it is nowhere said, which particular modified theory of gravity is used! And then, the process of obtaining its solution should be well described.

In the conclusion, it is said that solutions without horizons correspond to naked singularities. A few lines below, the absence of singularities is declared. What is true?

Lastly, second-order phase transitions are said to be found. Then, both in the text and in the abstract, it is necessary to say between which states there are such transitions, do they occur in the course of time (probably not since the solutions are static) or under other circumstances (which?).

It is clear that the paper cannot be published in its present form and may be
considered for publication only after major revision.

Author Response

Please see the attachment:

Reviewer 2 Report

Report is attached.

Author Response

Please see the attachment:

Reviewer 3 Report

Report to Universe-630294

In the manuscript, the author found non-singular black hole with nonlinear electrodynamics, and then they analyzed several thermodynamic properties including the Hawking temperature, heat capacity and phase transition. It would be interesting for part of the readers in this field. There are several issues that need to be addressed before the manuscript can be published.

The author should motivate their study, why they consider the Lagrangian Eq.(2) for the nonlinear gauge field? What is the physics? What is the case if one considers planar or hyperbolic horizon in Eq.(3)? In other words, are the conclusions still valid for general geometries of horizon? Can \beta in Eq.(2) be negative? If yes, B in Eq.(11) can also be negative, then what will happen of the temperature in figure.2.

Author Response

Please see the attachment:

Round 2

Reviewer 1 Report

Unfortunately, I still cannot recommend this paper for publication. The main reason is that if one takes the metric (3) with the Hayward function (4) and interprets it as a magnetic Einstein-NED solution, one immediately obtains the corresponding Lagrangian function L(F), and it will not coincide with (1). This can be obtained from the general relations given in [33]. Indeed, by definition, the mass function is defined by
$f(r) = 1 - 2m(r)/r,$
and comparing it with (4), we obtain
$\rho(r) = 6GM (r^3 + 6GM l^2)^{-2} = -L$

On the other hand, since for a magnetic field $F = 2q_m^2/r^4$, where $q_m$ is the magnetic charge, it is easy to express $L$ in terms of $F$, and it is $L(F) = - a (F^{-3/4} + b)^{-2}$
with constants $a, b$, quite different from (1). If this is not true, I would be grateful to the author for an explanation.

It would be good if the author avoided such wrong assertions as "M is the mass of a neutral BH" after (4) (whereas (4) is a metric function and gives no information on which kind of matter field, charged or neutral, is the source of gravity), and, before (5), "the BH mass varies with r"
(whereas the BH mass is a constant characterizing gravity as $r \to \infty$).

Other shortcomings are rather poor English and rather a chaotic introduction, though with a lot of references but with illogical jumps from one subject to another. One could expect from the introduction not only enumeration of "who studied what" but, mainly, why the author chooses this particular model and why (if it is so) it is better or more interesting than others.

So, in my view, in the present form the paper is not suitable for the journal, but I suggest the editors to give the author one more chance to improve it.

Author Response

Please see the attachment:

Reviewer 2 Report

Report is attached.

Author Response

Please see the attachment:

Reviewer 3 Report

The author clarified my considerations. I recommend the current version to be accepted for publication.

Author Response

Thanks.

Round 3

Reviewer 1 Report

This version of the paper, with a number of points corrected, is much better than both previous ones. Nevertheless, it needs some improvement of the presentation, which cannot be called deep reworking but rather a better and more correct explanation of different points. I'll enumerate my remarks as they refer to the text rather than in the order of their significance.

1. Pages 1,2, lines 4-7, we read: "Thus, solutions of the Einstein equations for neutral (the Schwarzschild metric), charged (the Reissner-Nordstrom metric) and rotated (the Kerr metric) black holes (BHs) have curvature singularities in the center (r = 0)."

It is incorrect: r=0 is really a center in the RN metric, while in the Schwarzschild metric it is in a nonstatic region and has a cosmological nature, and in the Kerr metric it has the shape of a ring.

2. Page 2, lines 28, 29: "In the papers [12], [13] the regular electrically charged BH solution in the GR was presented".

To begin with, these are different solutions rather than one solution. But much more importantly, as shown and discussed in detail in [28], in these and all other regular electric solutions there is a significant shortcoming: they need different NED theories in different parts of space. This must be
mentioned here; and then, it remains quite unclear what the author means by saying that "this NED satisfies the correspondence principle. " (line 31).

3. Page 3, lines 60-62. "In the present study we use NED of [28], work beyond the GR within the approach of [29], [30], and investigate the magnetically charged BH."

I think, it is insufficient here to refer to [29, 30], but it must be clearly said that the author uses a phenomenological extension of GR by formally introducing a fundamental length by using Hayward's metric (even though this is mentioned in the abstract).

4. Below Eq. (4) it is said that "the line element (3) with the metric function (4) corresponds to uncharged non-singular BH."

First, at large enough "l" there are no horizons, and it is not a BH. More important: in GR such a function may be ascribed to a solution with NED and a nonzero charge. It must be clearly said that it is the author's interpretation that this metric is ascribed to an extengion of GR, and it is introduced for an uncharged source to replace the Schwarzschild metric. I can add that, in the framework of GR, it would be meaningless to introduce a nonzero m_0 since the mass function for a solution with a regular center is specified by Eq.(5) with m_0 =0.

5. On line 88 the author mentions an ADM mass, but earlier he speaks of a Schwarzschild mass. To my knowledge, these notions coincide for an asymptotically flat metric in GR. If they differ in the author's extension of GR, it must be discussed - or a mention of ADM mass may be omitted.

6. Page 6, lines 120, 121 we read "that in the Einstein theory of gravity (l = 0) based on NED (1), singularities are still present."

However, according to [28], the magnetic GR solution with NED (1) is regular at r=0. So there is something wrong with the transition to l=0.

If all these points are corrected, then, in my opinion, the paper may be published.

Author Response

Report 1

This version of the paper, with a number of points corrected, is much better than both previous ones. Nevertheless, it needs some improvement of the presentation, which cannot be called deep reworking but rather a better and more correct explanation of different points. I'll enumerate my remarks as they refer to the text rather than in the order of their significance.

Q1:Pages 1,2, lines 4-7, we read: "Thus, solutions of the Einstein equations for neutral (the Schwarzschild metric), charged (the Reissner-Nordstrom metric) and rotated (the Kerr metric) black holes (BHs) have curvature singularities in the center (r = 0)."It is incorrect: r=0 is really a center in the RN metric, while in the Schwarzschild metric it is in a nonstatic region and has a cosmological nature, and in the Kerr metric it has the shape of a ring.

R1:The sentence was changed to be “Thus, solutions of the Einstein equations for charged (the Reisner-Nordstrom metric) black holes (BHs) have curvature singularities in the center (r=0).”

Q2:Page 2, lines 28, 29: "In the papers [12], [13] the regular electrically charged BH solution in the GR was presented". To begin with, these are different solutions rather than one solution. But much more importantly, as shown and discussed in detail in [28], in these and all other regular electric solutions there is a significant shortcoming: they need different NED theories in different parts of space. This must be mentioned here; and then, it remains quite unclear what the author means by saying that "this NED satisfies the correspondence principle. " (line 31).

R2:It was added (page 2): “It worth noting that in accordance with \cite{ Bronnikov} regular electric solutions, with the Maxwell weak-field limit, can be described only by different NED theories in different parts of spacetime. Thus, there is a significant shortcoming in the models of [12], [13].” The “correspondence principle” was removed (but I meant that correspondence principle is the Maxwell weak-field limit).

Q3:Page 3, lines 60-62. "In the present study we use NED of [28], work beyond the GR within the approach of [29], [30], and investigate the magnetically charged BH." I think, it is insufficient here to refer to [29, 30], but it must be clearly said that the author uses a phenomenological extension of GR by formally introducing a fundamental length by using Hayward's metric (even though this is mentioned in the abstract).

R3:I have modified the sentence: “In the present study we use NED of \cite{Bronnikov}, explore a phenomenological extension of GR by introducing a fundamental length $l$, using the Hayward's metric, and investigate the magnetically charged BH.”

Q4:Below Eq. (4) it is said that "the line element (3) with the metric function (4) corresponds to uncharged non-singular BH."First, at large enough "l" there are no horizons, and it is not a BH. More important: in GR such a function may be ascribed to a solution with NED and a nonzero charge. It must be clearly said that it is the author's interpretation that this metric is ascribed to an extension of GR, and it is introduced for an uncharged source to replace the Schwarzschild metric. I can add that, in the framework of GR, it would be meaningless to introduce a nonzero m_0 since the mass function for a solution with a regular center is specified by Eq.(5) with m_0 =0.

R4:It was added: “We interpret this metric, in the framework of an extension of GR, for an uncharged source and replace the Schwarzschild metric.” and “It should be noted that in GR the metric function (4) may be obtained as a solution within NED with a nonzero magnetic charge \cite{Fan}. But in this case the NED  Lagrangian is ill-defined.  At the weak-field limit the NED Lagrangian does not approach to the Maxwell Lagrangian.”

Ref. \cite{Fan} was added. The sentence: “Then the line element (3) with the metric function (4) corresponds to uncharged non-singular BH.” was removed.

Q5:On line 88 the author mentions an ADM mass, but earlier he speaks of a Schwarzschild mass. To my knowledge, these notions coincide for an asymptotically flat metric in GR. If they differ in the author's extension of GR, it must be discussed - or a mention of ADM mass may be omitted.

R5:A mention of ADM mass was omitted.

Q6:Page 6, lines 120, 121 we read "that in the Einstein theory of gravity (l = 0) based on NED (1), singularities are still present." However, according to [28], the magnetic GR solution with NED (1) is regular at r=0. So there is something wrong with the transition to l=0.

R6:The sentence “It is seen from Eqs. (16) and (18) that in the Einstein theory of gravity ($l=0$) based on NED (1), singularities are still present.” was omitted.

 If all these points are corrected, then, in my opinion, the paper may be published.

Reviewer 2 Report

I submitted my report on the revised version of this manuscript on 14 Nov 2019. For a case I attach this report here. Author's remarks over my report in his reply only confirm my evaluation of the level of this paper. I might add that in all my activity as a reviewer for leading journals (for decades) this is the first case when I determinantly recommend to reject the paper because of incorrect approach and incredible mistakes in logic and solution of the Einstein equations, as well as highly improper way of quotation of related papers presented in the literature.

Author Response

I submitted my report on the revised version of this manuscript on 14 Nov 2019. For a case I attach this report here.

I replied for all comments of 14 Nov. 2019 Report.

Q1:Author's remarks over my report in his reply only confirm my evaluation of the level of this paper. I might add that in all my activity as a reviewer case when I determinantly recommend to reject the for leading journals (for decades) this is the first case when I determinantly recommend to reject the paper because of incorrect approach and incredible mistakes in logic and solution of the Einstein equations,

R1:As was written in Abstract and in Introduction the  approach used is beyond the Einstein Gravity. In Abstract it was written: When the fundamental length introduced, characterising quantum gravity effects, vanishes one comes to the general relativity...". Only one misprint was found in Introduction. No mistakes in calculations were found. The statement: "incorrect approach and incredible mistakes in logic and solution of the Einstein equations" is absolutly wrong.   

Q2:as well as highly improper way of quotation of related papers presented in the literature.

R2:A lot of papers Referee 2 suggested to cite are not related to the present paper.